# C-STS: Conditional Semantic Textual Similarity

**Ameet Deshpande**[* 1,2]     **Carlos E. Jimenez**[* 1]     **Howard Chen**[1]
**Vishvak Murahari**[1]     **Victoria Graf**[1]     **Tanmay Rajpurohit**[3]
**Ashwin Kalyan**[2]     **Danqi Chen**[1]     **Karthik Narasimhan**[1]

[1] Princeton University     [2] The Allen Institute for AI     [3] Georgia Tech
asd@cs.princeton.edu

## Abstract

Semantic textual similarity (STS), a cornerstone task in NLP, measures the degree of similarity between a pair of sentences, and has broad application in fields such as information retrieval and natural language understanding. However, sentence similarity can be inherently ambiguous, depending on the specific aspect of interest. We resolve this ambiguity by proposing a novel task called Conditional STS (C-STS) which measures sentences' similarity conditioned on an feature described in natural language (hereon, *condition*). As an example, the similarity between the sentences *"The NBA player shoots a three-pointer."* and *"A man throws a tennis ball into the air to serve."* is higher for the condition *"The motion of the ball"* (both upward) and lower for *"The size of the ball"* (one large and one small). C-STS's advantages are two-fold: (1) it reduces the subjectivity and ambiguity of STS and (2) enables fine-grained language model evaluation through diverse natural language conditions. We put several state-of-the-art models to the test, and even those performing well on STS (e.g. SimCSE, Flan-T5, and GPT-4) find CSTS challenging; all yielding Spearman correlation scores below 50. To encourage a more comprehensive evaluation of semantic similarity and natural language understanding, we make nearly 19K C-STS examples and code available for others to train and test their models. [1]

## 1 Introduction

Over the years, natural language processing (NLP) has progressed through the co-evolution of model design (e.g. architectures, training methods) and evaluation methods for language tasks (Wang et al., 2018, 2019; Hendrycks et al., 2021). A common task used to evaluate NLP models has been Semantic Textual Similarity (STS) (Agirre et al., 2012), which evaluates the models' ability to predict the

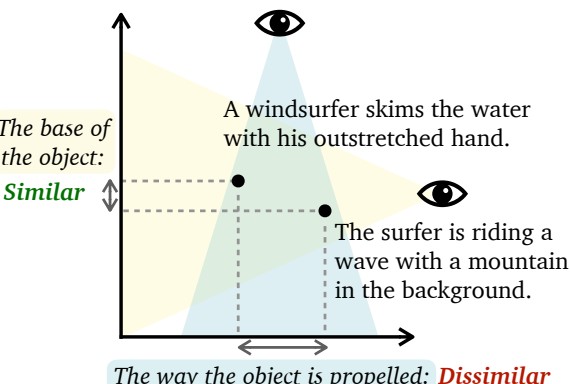

Figure 1: C-STS: Two sentences are judged by their similarities based on free-form natural language conditions. The two sentences are more similar when judged by the condition '*The base of the object*' (highlighted in yellow) as both windsurfing and surfing use a similar board but are dissimilar when judged by the condition '*The way the object is propelled*' (highlighted in blue) because one is propelled by waves and the other by wind. Providing conditions reduces ambiguity of the sentence similarity task, and allows evaluation of a grounded and multi-faceted notion of sentence similarity.

semantic similarity between two sentences. Several diverse STS datasets are popularly used, with prior work expanding the STS task to multiple domains and languages (Agirre et al., 2013, 2014, 2015, 2016; Cer et al., 2017; Abdalla et al., 2021). STS tasks have been a component of the popular GLUE natural language understanding benchmark (Wang et al., 2018) and are a key evaluation tool for sentence-representation learning specifically (Conneau et al., 2017; Cer et al., 2018; Reimers and Gurevych, 2019; Gao et al., 2021, inter alia).

Despite its popularity, STS may be inherently ill-defined. The general semantic similarity of two sentences can be highly subjective and vary wildly depending on which aspects one decides to focus on. As observed in several studies, ambiguity in similarity judgements of word or sentence pairs can be reduced with the help of context for both hu-

---

[*] Equal Contribution

[1]Code: www.github.com/princeton-nlp/c-sts

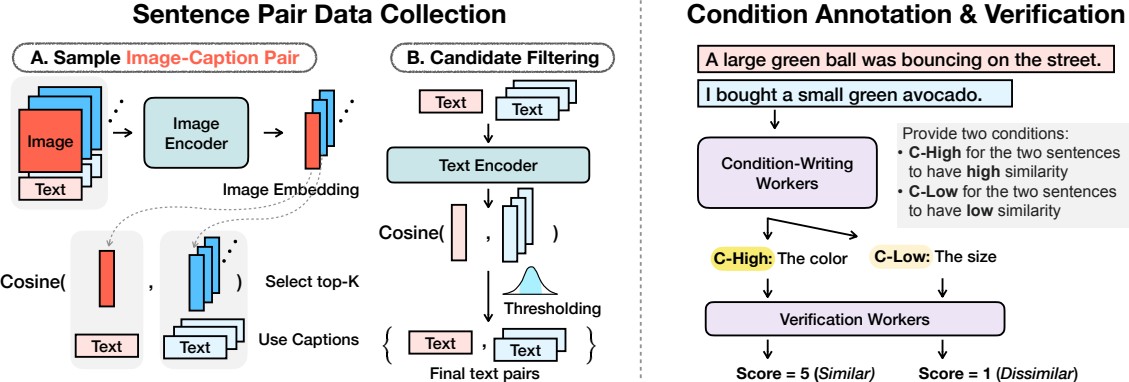

Figure 2: Illustrating the data collection process for C-STS-2023. (Left) We first show the sentence pair collection procedure (§2.2.1). Step A: An image-caption pair is sampled (red) from the dataset and then fed into the image encoder to get the image embedding. The image embedding is compared against all other image embeddings in the dataset (blue) to find the top-$k$ similar images. The original caption is then paired with the corresponding captions of the top-$k$ similar images to generate sentence pairs. Step B: The sentence pairs are filtered based on textual similarity. (Right) We illustrate the condition annotation/verification procedure (§2.2.2). Once the sentence pairs have been collected, they are sent to qualified Mechanical Turkers to get annotations and verify conditions.

mans (De Deyne et al., 2016a,b) and models (Veit et al., 2016; Ye et al., 2022a; Lopez-Gazpio et al., 2017; Camburu et al., 2018).

Considering the importance of STS tasks for evaluating sentence representations, we propose a new task called *Conditional* STS (C-STS), illustrated in Figure 1, which seeks to disambiguate the similarity of two sentences by measuring similarity within the context of a condition sentence.

C-STS, uses free-form natural language conditions, enabling us to evaluate and probe natural language understanding for myriad fine-grained aspects. Figure 1 illustrates two conditions ("*The base of the object*" and "*The way the object is propelled*") which probes language models' conception of similarity for different aspects concerning water sports and physical reasoning. Since conditions themselves are unconstrained sentences, they allow us to evaluate a precise, grounded, and multi-faceted notion of sentence similarity.

To comprehensively test models on C-STS, we create the C-STS-2023 dataset which includes 18,908 instances containing sentence pairs, a condition, and a scalar similarity judgement on the Likert scale (Likert, 1932). We find that even state-of-the-art sentence encoders and large language models perform poorly on our task. Although Sim-CSE (Gao et al., 2021) and GPT-4 (OpenAI, 2023a) are among the best-performing systems, their relatively poor Spearman correlation of 47.5 and 43.6 respectively, points to significant room for improvement (SimCSE achieves a Spearman correlation of 88.09 on STS-B validation splits for comparison).

We believe that C-STS provides a testbed for potentially novel modeling settings and applications. Toward this end, we propose and evaluate a unique encoding setting (a tri-encoder) and objectives (a quadruplet contrastive loss with hard negatives) that take advantage of C-STS's three-sentence inputs and paired high- and low-similarity instances.

Our qualitative analysis shows that models find C-STS challenging when tested on different aspects of the same sentence pair rather than testing an unconditional and ambiguous notion of similarity. We hope that future work evaluates on C-STS in addition to STS tasks to comprehensively benchmark semantic similarity in language models.

## 2 Methodology

The C-STS task requires sentence pairs, conditions which probe different aspects of similarity, and the similarity label for a given sentence pair and condition. This section describes the technical details involved in creating the dataset.

### 2.1 Background: Semantic textual similarity

Semantic textual similarity (STS) (Agirre et al., 2012, 2013, 2014, 2015, 2016; Cer et al., 2017) is a task which requires machines to make similarity judgements between a pair of sentences ($\{s_1, s_2\}$). STS measures the unconditional semantic similarity between sentences because the annotator making the similarity assessment must *infer* which as-

pect(s) of the sentences are being referred to. Formally, consider conditions ($c_i \in \mathcal{C}$) that refer to disjoint aspects of the sentences, then the similarity of the two sentences may be represented as:

$$\sum_{i=1}^{|\mathcal{C}|} w_i \, \text{sim}_{c_i} (s_1, s_2) \quad s.t. \sum_i w_i = 1$$

Here, $w_i$ is the weight assigned by the annotator to the condition $c_i$, and $\text{sim}_{c_i} (s_1, s_2)$ is the similarity of the sentences with respect to the condition. These weights are latent to the task and each annotator has their own interpretation of them which helps marginalize similarity, thus introducing ambiguity in the task. C-STS seeks to disambiguate the STS task by measuring similarity conditioned by a single aspect specified in natural language.

## 2.2 Conditional semantic textual similarity

C-STS is a task comprised of quadruplets containing two sentences (a sentence pair), a natural language condition, and a similarity assessment ($\{s_1, s_2, c, y\}$). Crucially, we do not place any strict constraints on $c$, allowing it to be any relevant phrase. This allows us to probe potentially any possible aspect of similarity that may be considered between sentences.

### 2.2.1 Sentence Data Collection

The first stage of making the C-STS dataset is to acquire the sentence pairs that will later be used in eliciting conditioning statements from annotators.

We source sentence pairs $\{s_1, s_2\}$ for our dataset from image-captioning datasets through a two-step process: (1) generate candidate text pairs through dense retrieval from the corresponding *image representations* and (2) filter out candidates that are irrelevant or ineffectual for our purposes.

**Image Retrieval** Image-captioning datasets provide a compelling data source because image pair similarity and caption (text) pair similarity encode different semantics (Parekh et al., 2021). Image-representations thus serve as an informative latent variable which can represent their captions in ways that are not captured by text retrievers.

Since current sentence representation models overlook aspects of conditional similarity, we utilize both the image and text to retrieve sentence pairs which form the foundation of our dataset.

We aim to derive sentence pairs from an image-caption dataset $\mathcal{D}$ to aid in creating conditioning statements. To do this, we first generate a store of image pairs, or $\mathcal{P}_I$. Each pair, denoted by $I_i, I_j$, is such that $I_j$ is amongst the top-$k$ most similar images to $I_i$, determined by the cosine distance metric of their respective image representations obtained via an image encoder $E_I(\cdot)$. After establishing $\mathcal{P}_I$, we convert it into a sentence pair store ($\mathcal{P}_S$) by replacing each image in a pair with its corresponding caption. When each image $I_i \in \mathcal{D}$ is associated with a set of sentences $\{s\}_i$ we take all sentence pairs from the Cartesian product $\{s\}_i \times \{s\}_j$ for each image pair $I_i, I_j \in \mathcal{P}_I$.

**Candidate Filtering** After acquiring initial sentence pairs through image retrieval, we perform additional filtering to eliminate sentence pairs which are ill-suited for our task.

Specifically, we aim to include only pairs of sentences for which the unconditional similarity is somewhat ambiguous, as this incentivizes models to rely on the condition when reasoning about the conditional similarity.

To this end, we avoid high similarity sentence pairs by filtering out those with a high bag-of-words intersection over union and avoid low similarity sentence by choosing sentences with moderate or high cosine similarity of their SimCSE embeddings (Gao et al., 2021). See Appendix A.2 for a full description of all filtering criteria used.

**Dataset sources** For the construction of sentence pairs candidates, we use two image-caption datasets: the train split from the 2014 MS COCO dataset (Lin et al., 2014) containing $\sim 83,000$ images, and Flickr30K (Young et al., 2014) containing $\sim 31,000$ images. Each dataset is processed separately and we do not intermix them during the retrieval stage. We use CLIP-ViT (Radford et al., 2021) to encode images and include the specific filtering criteria in Table 6 of Appendix A.2.

### 2.2.2 Annotation Methodology

For each sentence pair in the store ($\mathcal{P}_S$), we wish to collect conditions and semantic similarity annotations for each sentence pair and condition triplet, $\{s_1, s_2, c\}$. As $c$ is a free-form natural language sentence, the annotator is provided with a high-level of control over which aspect to condition on. Human annotations are acquired through Mechanical Turk in a 3-stage process.

**Stage 1: Choosing a high-quality worker pool** In the first stage, we design a qualification test to select workers who excel at our task. Specifically, we test two skills: (1) The quality of conditions they

write for a given sentence pair and (2) semantic similarity judgements for a triplet $\{s_1, s_2, c\}$. We choose a pool of 271 workers who perform well on both tasks and restrict subsequent stages to include only workers from this pool. See Appendices C.1 and C.2 for an example of these tests.

**Stage 2: Condition annotation** After sourcing sentence pairs $\{s_1, s_2\}$ using the strategy discussed in the Section 2.2.1, we instruct workers to annotate each pair with one condition such that the similarity in its context is high (C-High) and one such that the similarity in its context is low (C-Low). Example:

**s1** : A large green ball was bouncing on the street

**s2** : I bought a small green avocado

**C-High** : The color of the object

**C-Low** : The size of the object

We do not place any constraints on the conditions other than that they should be semantically unambiguous phrases and relevant to the sentence pair (Appendix C.1).

**Stage 3: Condition verification and similarity assessment** The output of annotations from the previous stage are triplets $\{s_1, s_2, c\}$ with a binary similarity assessment (high or low). In this stage we ask new annotators to assign a similarity on a Likert scale (Likert, 1932) (as an integer between 1 and 5) as is common with semantic textual similarity tasks (Agirre et al., 2012). In addition to assigning a similarity, we also use this stage to verify if the conditions from the previous stage are pertinent to the sentence pairs, filtering out potentially low quality examples. At the end of this stage, we have $\{s_1, s_2, c, y\}$ quadruplets which have passed a layer of human verification (Appendix C.2).

## 3 Dataset Analysis

**Dataset statistics** To ensure high-quality, faithful, and diverse annotations, we collect a total of 20,000 instances and perform quality assurance (Section 5.3) resulting in a total of 18,908 instances as part of the C-STS-2023 dataset. Following standard practice, we create train, validation, and test splits in a $60 : 15 : 25$ ratio. We present the distribution of similarity scores, which are discrete numbers between $[1, 5]$, in Figure 4. We also measure the inter-annotator agreement on a random sample of 100 examples with three independent annotations and find Fleiss' kappa score (Fleiss,

1971) to be 0.61 which implies substantial inter-annotator agreement. Average length of sentences and conditions is 12.6 and 5.3 words.

**Qualitative analysis** C-STS allows us to evaluate the generally fuzzy notion of sentence similarity with greater fidelity. We illustrate this in Table 1, where precise and discriminative conditions allow a targeted, fine-grained, and grounded definition of sentence similarity. The following is a representative instance where the conditions tease out nuanced and hidden similarities and differences between the two lexically similar sentences on surfing: Consider $s_1$: "*A windsurfer skims the water...*" and $s_2$: "*The surfer is riding a wave...*"). While the sentences are significantly dissimilar based on the condition "*the way the object is propelled*" as they talk about windsurfing and surfing respectively (the former uses a sail whereas the latter depends on the wave), they are very similar in context of the condition "*the base of the object*" as both windsurfing and surfing use a similar board.

Our diverse set of conditions provides broad support over the distribution of conditions and enables a holistic and multi-faceted evaluation of sentence similarity. For example, the conditions for the sentences on Tennis in Table 1 test similarity both on the sport being played (which requires understanding lexical and knowledge artifacts) as well as the number of people (which requires reasoning and commonsense capabilities).

## 4 Baselines

We evaluate our dataset on several baselines which can be categorized into (1) Fine-tuning baselines, which are pre-trained models finetuned on the C-STS training split, and (2) Large language models (LLMs) baselines, which are evaluated using instructions and in-context examples.

### 4.1 Fine-tuning baselines

We evaluate three sentence encoder models RoBERTa (Liu et al., 2019), supervised SimCSE (Gao et al., 2021) and unsupervised DiffCSE (Chuang et al., 2022). SimCSE and DiffCSE represent state-of-the-art sentence encoder models which are particularly strong on STS tasks. For both SimCSE and DiffCSE, we use the RoBERTa pre-trained varieties.

**Encoding configurations** Encoder-only Transformer models, such as BERT (Devlin et al., 2019) and RoBERTa (Liu et al., 2019), initially performed

| Sentence 1 | Sentence 2 | Condition and Similarity |
|---|---|---|
| An older man holding a glass of wine while standing between two beautiful ladies. | A group of people gather around a table with bottles and glasses of wine. | *The people's demeanor*: 5 
 *The number of bottles*: 1 |
| Various items are spread out on the floor, like a bag has been emptied. | A woman with a bag and its contents placed out before her on a bed. | *The arrangement of objects*: 4 
 *The surface the objects are on*: 1 |
| A windsurfer skims the water with his outstretched hand. | The surfer is riding a wave with a mountain in the background. | *The base of the object*: 5 
 *The way the object is propelled*: 1 |
| Female tennis player jumping off the ground and swinging racket in front of an audience | A young lady dressed in white playing tennis while the ball girl retrieves a tennis ball behind her. | *The sport being played*: 5 
 *The number of people*: 1 |

Table 1: Four examples from the C-STS validation set. Under different conditions, the same sentence pair can be separated into high similarity and low similarity. Scale from 1 (dissimilar) to 5 (similar).

regression finetuning for STS tasks by simply concatenating the sentences and encoding them together before generating a prediction; let us call this type of architecture a *cross-encoder*. Recent approaches instead opt to encode sentences separately and compare their similarity using a distance metric, such as the cosine distance Reimers and Gurevych (2019); which we will call a *bi-encoder*. While DiffCSE and SimCSE were designed with the bi-encoder setting in mind, we observe that they work well in the cross-encoder setting as well.

For our baselines, we evaluate each model in both settings. For the cross-encoder configuration, we encode the triplet containing the sentences and the condition ($\{s_1, s_2, c\}$), and the output is a scalar similarity score − $f_\theta(s_1; s_2; c)$. For the bi-encoder configuration (Reimers and Gurevych, 2019), the sentences of a pair are encoded independently along with the condition using a Siamese network and their cosine similarity is computed − $\cos(f_\theta(s_1; c), f_\theta(s_2; c))$.

In addition to the bi- and cross-encoder models, we propose *tri-encoder* models which encode each sentence and condition separately. This conceptually resembles late-interaction contextualized retrieval approaches, such as Humeau et al. (2020) or Khattab and Zaharia (2020), but our approach is specific to C-STS. For this, we first encode all sentences of the triplet separately, with encoder $f_\theta(\cdot)$ as $\mathbf{s}_i = f_\theta(s_i)$, where $\mathbf{s}_i \in \mathbb{R}^d$. We then perform an additional transformation $h : \mathbb{R}^{2d} \to \mathbb{R}^d$ that operates on the condition and one each of the sentences. We finally compute the conditional similarity using

the cosine similarity as $\cos(h(\mathbf{c}; \mathbf{s}_1), h(\mathbf{c}; \mathbf{s}_2))$. We experiment with 2 functions for $h$, an MLP and the Hadamard product.

**Objectives** In addition to the standard MSE loss for regression, we use a quadruplet contrastive margin loss which we denote Quad. Since each sentence pair in C-STS comes with two conditions (one with higher similarity and one with lower similarity) we represent the conditional encoding of each sentence in the higher-similarity pair as $p_1$ and $p_2$ and represent the conditional encoding of each sentence in the lower similarity pair as $n_1$ and $n_2$. The Quad loss is then defined as follows:

$$\mathrm{Quad}(p_1, p_2, n_1, n_2) = \max(\lambda + \cos(n_1, n_2) \\ - \cos(p_1, p_2), 0)$$

where $\lambda$ is a margin hyperparameter.

We train all of our tasks for regression using, alternatively, mean squared error (MSE), Quad, and a linear combination of the quadruplet loss and MSE (Quad + MSE). Since we require a separate conditional encoding fore each sentence, the Quad and (Quad + MSE) objectives apply only the the bi-encoder and tri-encoder configurations.

**Hyperparameters** We evaluate the baselines on the test split for C-STS. We perform a hyperparameter sweep to select the best performing configuration and test using models trained with 3 random seeds, with further details in Appendix A.3. As a comparison for our training setting, we perform a similar hyperparameter sweep for the STS-B (Cer et al., 2017) dataset, with the validation split results

and best hyperparameters shown in Table 9, showing that our finetuned baselines achieve very strong performance on traditional STS tasks.

## 4.2 Large language models baselines

For the generative setting, we evaluate two types of models (1) instruction-finetuned encoder-decoder models, including Flan-T5 (Chung et al., 2022), Flan-UL2 (Tay et al., 2023), and Tk-INSTRUCT (Wang et al., 2022) and (2) proprietary autoregressive LLMs including ChatGPT-3.5 (OpenAI, 2022) and GPT-4 (OpenAI, 2023a). For ChatGPT-3.5 and GPT-4, we use the OpenAI API with versions `gpt-3.5-turbo-0301` and `gpt-4-0314` respectively.

When evaluating zero- or few-shot capabilities, each model input is composed of up to three parts: instruction (task definition), $k$ in-context examples, and query. Models are evaluated with $0, 2$, or $4$ examples and using three different instruction prompts: no instruction, short instruction, which provides only a high-level description of the task, and long instruction, shown in Figure 6, which resembles the annotation guidelines and is similar to the instructions used for the STS-B classification task in Wang et al. (2022).

For few-shot evaluation, we additionally always group a sentence pairs' two conditional similarity examples together, so models will always see contrasting pairs in the examples, but won't see a paired example for the query. We provide examples of the formats used for the input and output for more settings in Appendix B. As we did for the finetuned models, we also evaluate these models on the STS-B validation split, shown in Table 12, with instruction finetuned models and ChatGPT achieving strong performance.

## 5 Results

### 5.1 Evaluating sentence encoders on C-STS

**Zero-shot bi-encoder performance** As an initial comparison, we evaluate bi-encoder models without finetuning, on both C-STS and STS-B. As shown in Table 2, we see that strong performance on STS-B does not translate to good performance on C-STS, suggesting that these models fail entirely to incorporate the provided conditioning statement. These results suggest that current approaches to training sentence encoders may be too specialized to existing tasks for evaluation, such as STS-B.

| Model | C-STS | | STS-B | |
|---|---|---|---|---|
| | Spear. | Pears. | Spear. | Pears. |
| DiffCSE$_{\text{BASE}}$ | 0.9 | 0.5 | 84.4 | 85.1 |
| RoBERTa$_{\text{BASE}}$ | -0.4 | -0.1 | 35.2 | 48.2 |
| RoBERTa$_{\text{LARGE}}$ | -1.8 | -2.4 | 7.3 | 15.1 |
| SimCSE$_{\text{BASE}}$ | 1.7 | 0.8 | 85.1 | 86.8 |
| SimCSE$_{\text{LARGE}}$ | **1.9** | **1.4** | **88.1** | **88.9** |

Table 2: Zero-shot bi-encoder models evaluation results on C-STS and STS-B validation data. These results verify that strong performance on STS tasks do not translate to C-STS, suggesting substantial room for improvement for fine-grained sentence embedding models.

| Encoding | Model | Spear.↑ | Pears.↑ |
|---|---|---|---|
| Cross-encoder | RoBERTa$_{\text{BASE}}$ | $39.2_{\pm1.3}$ | $39.3_{\pm1.3}$ |
| | RoBERTa$_{\text{LARGE}}$ | $40.7_{\pm0.5}$ | $40.8_{\pm0.4}$ |
| | DiffCSE$_{\text{BASE}}$ | $38.8_{\pm2.9}$ | $39.0_{\pm2.7}$ |
| | SimCSE$_{\text{BASE}}$ | $38.6_{\pm1.3}$ | $38.9_{\pm1.2}$ |
| | SimCSE$_{\text{LARGE}}$ | $\mathbf{43.2}_{\pm1.2}$ | $\mathbf{43.2}_{\pm1.3}$ |
| Bi-encoder | RoBERTa$_{\text{BASE}}$ | $28.1_{\pm8.5}$ | $22.3_{\pm14.1}$ |
| | RoBERTa$_{\text{LARGE}}$ | $27.4_{\pm6.2}$ | $21.3_{\pm8.4}$ |
| | DiffCSE$_{\text{BASE}}$ | $43.4_{\pm0.2}$ | $43.5_{\pm0.2}$ |
| | SimCSE$_{\text{BASE}}$ | $44.8_{\pm0.3}$ | $44.9_{\pm0.3}$ |
| | SimCSE$_{\text{LARGE}}$ | $\mathbf{47.5}_{\pm0.1}$ | $\mathbf{47.6}_{\pm0.1}$ |
| Tri-encoder | RoBERTa$_{\text{BASE}}$ | $28.0_{\pm0.4}$ | $25.2_{\pm1.0}$ |
| | RoBERTa$_{\text{LARGE}}$ | $20.3_{\pm2.2}$ | $18.9_{\pm2.3}$ |
| | DiffCSE$_{\text{BASE}}$ | $28.9_{\pm0.8}$ | $27.8_{\pm1.2}$ |
| | SimCSE$_{\text{BASE}}$ | $31.5_{\pm0.5}$ | $31.0_{\pm0.5}$ |
| | SimCSE$_{\text{LARGE}}$ | $\mathbf{35.3}_{\pm1.0}$ | $\mathbf{35.6}_{\pm0.9}$ |

Table 3: We report fine-tuned model test split results in Spearman and Pearson correlations for three models (RoBERTa, DiffCSE, and SimCSE) in different encoding settings.

**Fine-tuning baselines** We finetune our sentence encoder baselines on C-STS and show the test performance in Table 3. Again, the best models are SimCSE and DiffCSE in the bi-encoding setting. This is suggests that the sentence representations learned in their contrastive learning phase facilitate learning for C-STS substantially, but still struggle with all Spearman correlation below 50.

Performance on C-STS varies significantly depending on the encoding configurations, with the bi-encoder setting proving to be the most effective, especially for SimCSE and DiffCSE models. Performance of the tri-encoder model, introduced in Section 4.1 was generally poor, with all models performing well below their bi-encoding and cross-encoding counterparts.

| Instruct. | Model | 0-shot↑ | 2-shot↑ | 4-shot↑ |
|---|---|---|---|---|
| | †SimCSE$_{\text{LARGE}}$ | | 47.5 | |
| None | Flan-T5$_{\text{XL}}$ | 1.7 | 11.3 | 16.4 |
| | Flan-T5$_{\text{XXL}}$ | 5.6 | 10.1 | 12.8 |
| | Flan-UL2 | 5.1 | 18.8 | 14.9 |
| | Tk-*Instruct*$_{\text{3B}}$ | -1.8 | 1.1 | 2.8 |
| | Tk-*Instruct*$_{\text{11B}}$ | 5.6 | 4.3 | 4.4 |
| | GPT-3.5 | 12.6 | 1.6 | 3.1 |
| | GPT-4 | **21.0** | **18.7** | **27.0** |
| Short | Flan-T5$_{\text{XL}}$ | 24.7 | 25.3 | 24.8 |
| | Flan-T5$_{\text{XXL}}$ | 30.6 | 29.7 | 29.2 |
| | Flan-UL2 | 20.7 | 22.4 | 23.2 |
| | Tk-*Instruct*$_{\text{3B}}$ | -0.3 | 3.9 | 4.9 |
| | Tk-*Instruct*$_{\text{11B}}$ | 10.1 | 21.9 | 17.1 |
| | GPT-3.5 | 15.0 | 15.6 | 15.5 |
| | GPT-4 | **39.3** | **42.6** | **43.6** |
| Long | Flan-T5$_{\text{XL}}$ | 26.6 | 26.3 | 26.0 |
| | Flan-T5$_{\text{XXL}}$ | 30.5 | 30.1 | 30.6 |
| | Flan-UL2 | 21.7 | 22.9 | 23.5 |
| | Tk-*Instruct*$_{\text{3B}}$ | -0.9 | 3.9 | 3.9 |
| | Tk-*Instruct*$_{\text{11B}}$ | 12.0 | 20.7 | 17.8 |
| | GPT-3.5 | 9.9 | 16.6 | 15.3 |
| | GPT-4 | **32.5** | **41.8** | **43.1** |

Table 4: Few-shot Spearman correlation on the test split. Models perform much worse than their finetuned counterparts, with GPT-4 being the only evaluated model that achieves comparable performance to some fine-tuned baselines. †: Fine-tuning on the full train set.

## 5.2 Evaluating pre-trained LLMs

We show performance of generative models evaluated on C-STS in various prompting settings in Table 4, with some additional results for smaller Flan-T5 models in Table 11 in the Appendix. Notably, the state-of-the-art language model, GPT-4, performs substantially better than all competing models and systems (UL2, Flan-T5, ChatGPT-3.5) and is competitive with a finetuned SimCSE$_{\text{LARGE}}$ model, the best performing sentence-encoder. For example, in most settings, GPT-4 outperforms ChatGPT-3.5 and Flan models by over 10 points. This suggests existing large language benchmarks may correlates with C-STS as GPT-4 has shown to be the most proficient in a wide variety of evaluation settings (OpenAI, 2023b).

Between suites of models of different sizes (viz. Flan-T5, Tk-*Instruct*), we observe a strong correlation between model scale and performance. We also find that providing instructions improves performance substantially for C-STS and that this performance is robust to different instructions lengths and the number of in-context examples.

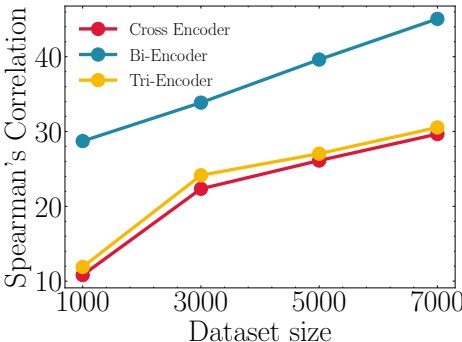

Figure 3: Model (SimCSE$_{\text{LARGE}}$) performance scaling as the dataset size increases. Across encoder types, Spearman correlation increases as the dataset scales.

## 5.3 Analysis

**Scaling laws for C-STS** We evaluate the effect of the quantity of C-STS data on sentence-embedding methods for SimCSE$_{\text{LARGE}}$ (Figure 3). We notice that for all three encoding strategies, performance monotonically increases as we increase the size of the training dataset. For example, for the SimCSE bi-encoder, the Spearman correlation increases from 30 when using a train set of 1,000 examples to 45 for 7,000 examples.

There is almost a linear increase in the performance of the models, especially the bi-encoder as we increase the amount of data. This quantitatively enforces the quality of the dataset, but also retroactively makes that point that rather than relying on more data, we require better modeling strategies.

**Qualitative analysis** We present predictions from different models in Table 5 to illustrate systematic pitfalls. For instance, Flan-T5 makes incorrect predictions even for straightforward instances and falsely predicts that both sentences talk about the same dish, even though the sentences clearly talk about sandwiches and pizza respectively. Additionally, ChatGPT-3.5 incorrectly predicts that the two sentences are completely dissimilar when talking about the types of plants, even though both sentences mention flowering plants. Note that our annotation, unlike ChatGPT-3.5, captures the nuance that the first sentence talks about *both shrubbery and flowers*, while the second sentence talks only about flowers, and therefore assigns a conservative similarity score of 3. The most proficient model on C-STS, GPT-4, is much better at capturing these nuances and accurately predicts, for instance, that the height of the giraffe's head (refer to the fourth example), is high in one sentence and

| Model | Sentence 1 | Sentence 2 | Condition | Output |
|---|---|---|---|---|
| Flan-T5-Base | A man taking a bite out of a sandwich at a table with someone else. | A man sitting with a pizza in his hand in front of pizza on the table. | Type of dish. | Pred: 4.5 Label: 1.0 |
| GPT-3.5 | A wooden bench surrounded by shrubbery and flowers on the side of a house. | A scene displays a vast array of flower pots in front of a decorated building. | The type of plants. | Pred: 0..0 Label: 3.0 |
| GPT-4 | Football player jumping to catch the ball with an empty stand behind him. | A football player preparing a football for a field goal kick, while his teammates can coach watch him. | The game being played. | Pred: 3.0 Label: 5.0 |
| GPT-4 | A giraffe reaches up his head on a ledge high up on a rock. | A giraffe in a zoo bending over the fence towards where impalas are grazing. | The height of the giraffe's head. | Pred: 1.0 Label: 1.0 |

Table 5: Examples of model predictions evaluated on C-STS in the in-context setting ($K = 2$ with no instructions). We choose examples with different levels of accuracy, showcasing different failure cases of model behavior.

low in another. GPT-4 is far from perfect though, and we outline a negative prediction (refer to the third example), where the model does not predict that the two sentences talk about the same game, even though they are very clearly about "Football".

More broadly, C-STS provides a lens into a model's ability to understand and reason over specific parts of each sentence and is well-suited to revealing systematic modeling issues.

## 6   Related Work

**Historical perspectives of semantic similarities** Measuring semantic similarities is a long-standing problem spanning cognitive science (Miller and Charles, 1991) to psychology (Tversky, 1977) where early attempts are made to quantify the subjective similarity judgements with information theoretical concepts. More recently, interest in semantic similarity has gained popularity in the context of machine learning, with works in computer vision recognizing that the notion of similarity between images varies with conditions (Veit et al., 2016) and can therefore be ambiguous (Ye et al., 2022b).

**Textual similarity tasks** Capturing textual similarity is also considered a fundamental problem in natural language processing. Works such as Agirre et al. (2012, 2016) define the textual semantic similarity tasks (STS), which is widely used in common benchmarks such as GLUE (Wang et al., 2018). Extensions to the STS setting have been proposed such as making the task broader with multilinguality (Cer et al., 2017) or incorporating relatedness (Abdalla et al., 2021). However, the loose definition of similarity has not been acknowledged as an issue explicitly. In contrast, our work tackles the

ambiguity problem by collecting conditions and hence reduce subjectivity. To alleviate ambiguity, explanations play an important role in identifying the differences between the two sentences either in their syntactical structure (Lopez-Gazpio et al., 2017) or in natural language (Camburu et al., 2018), but the post-hoc nature of explanations prevents it from being used prior to the similarity judgement, rendering it a supplemental component as opposed to a paradigm change in the task setup. Beyond STS, works that leverage conditioning to enhance sentence representations obtain improved performance for retrieval (Asai et al., 2023) and embedding qualities (He et al., 2015; Su et al., 2023; Jiang et al., 2022), which corroborates the observation that conditioning as a form of disambiguation benefits similarity measures.

## 7   Conclusion

In this work, we propose conditional semantic textual similarity (C-STS), a novel semantic similarity assessment task that resolves the inherent ambiguity in STS. Given the importance of STS and its importance in sentence representation evaluation we believe that C-STS is a timely and necessary addition to the language model evaluation landscape. Rather than testing unconditional semantic similarity, the diversity of conditions in our dataset allows fine-grained evaluation. The same sentence pairs can be tested on a variety of different aspects represented by conditions, with similarities often varying significantly. C-STS poses a challenging hurdle to both encoder-only and state-of-the-art generative language models which struggle to capture the high-dimensional manifold of similarity.

We believe that a combination of improved modeling and fine-tuning strategies are required to push the boundaries on C-STS and we hope that C-STS can enable innovative future work in language understanding and representation learning.

## Limitations

We propose the novel task of conditional semantic textual similarity (C-STS). Given that this is a new task, we collect a dataset of over $19,000$ instances, but one limitation that this size can be increased to ensure sentence embedding style models have additional data for fine-tuning. Further, we use two different sources to collect our sentence pairs, and future studies, motivated by STS follow-ups, can collect data from other sources.

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

## A Appendix

### A.1 Distribution of annotated similarity in the dataset

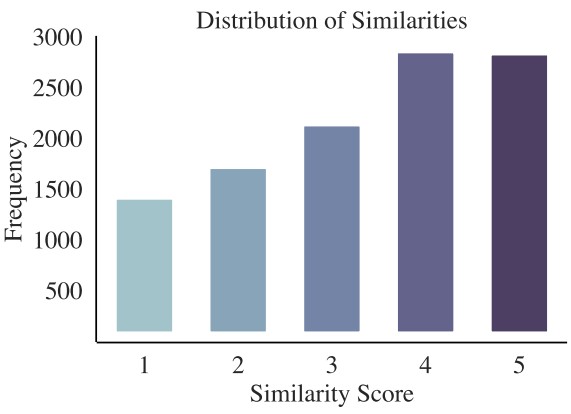

Figure 4: The train split distribution of similarity judgements on a Likert scale between $[1-5]$.

The distribution of similarities is equitably spread out over the Likert scale, as depicted in Figure 4.

### A.2 Sentence Pair Generation Details

Here we include some further details about sourcing sentence pairs from image-caption datasets.

As discussed in Section 2, we use a variety of metrics to quantitatively characterize the sentence pairs, and then to filter with the goal of removing pairs with excessively high or low unconditional similarity. The general criteria we consider are defined as follows:

- IOU - This is computed by taking the intersection over union of the bag of words for each sentence, after stopword removal. It represents the lexical similarity and overlap of a sentence pair.

- $d_{\text{text}}$ - The cosine distance of the pair's SimCSE embeddings. We chose SimCSE due to its ubiquity and effectiveness.

- ratio - This is the ratio of the shorter sentence's word count to the longer sentence's word count in a given pair.

- length - This is the character length of the shortest sentence in a pair.

Using these criteria, we filter the sentence pairs based upon thresholds (exact values shown in Table 6) where sentences are *rejected* if they violate

any of these criteria. These thresholds were selected based primarily manual inspection of samples on their margins. Criteria such as ratio and length are used primarily to facilitate comparison. Sentences with very different lengths are more difficult to compare, as are sentences that are very short or contain few details.

|  | COCO | Flickr30K |
|---|---|---|
| $k$ | 64 | 128 |
| IOU | $\leq 0.12$ | $\leq 0.2$ |
| $d_{\text{text}}$ | $\geq 0.4$ | $\geq 0.4$ |
| ratio | $\geq 0.7$ | $\geq 0.7$ |
| length | $\geq 50$ | $\geq 48$ |

Table 6: The list of filters criteria and values used for each dataset. Sentence pairs that violate any criterion are discarded.

### A.3 Evaluation Details

**Implementation Details** All models, with the exception of the ChatGPT systems, are trained and or evaluated in PyTorch using the Huggingface Transformers library (Wolf et al., 2019) and pre-trained weights repository. We use the STS-B dataset as distributed on https://huggingface.co/docs/datasets as part of the GLUE (Wang et al., 2018) evaluation benchmark.

**Finetuned Baselines** For evaluation of the finetuned baselines on C-STS, we perform a hyperparameter sweep to select the best training settings for each model and encoding method before evaluating on the test split of C-STS. We show the hyperparameter values used in the sweep in Table 7, and the final hyperparameter values chosen in Table 8. We evaluate 3 random seeds using the best validation configuration to evaluate on the test data, with final results reported in Table 3.

We additionally perform an extensive evaluation of our models on STS-B. We perform a comparable validation sweep as shown in Table 7, reporting the best performing hyperparameters and their performance in Table 9.

Lastly, we perform a data ablation training a RoBERTa$_{\text{BASE}}$ model alternatively on only the condition and only the sentence pair. The model trained to predict similarity based on the condition statement alone recovers non-trivial performance, but falls well behind the full-input baseline.

| | |
|---|---|
| Batch Size | {32} |
| Encoding Type | {Cross, Bi-, Tri-} |
| Epochs | {3} |
| Learning Rate | {1e-5, 3e-5} |
| LR Schedule | {linear} |
| Objective | {MSE, Quad, Quad + MSE} |
| Pooler Type | {[CLS] w/ MLP, * w/o MLP} |
| Seed | {42} |
| Warmup Ratio | {0.1} |
| Weight Decay | {0, 0.1} |

Table 7: Hyperparameter sweep done for C-STS validation for finetuned models. The same sweep, with the exception of the Encoding Type and Objective hyper parameters are done for STS-B.

**Generative Baselines** We report more details of results of the generative baselines for the validation sets of C-STS and STS-B.

For comparison to validation performance of other models, we include the validation performance for C-STS in Table 11, which largely mirrors performance on the test set. We notice, expectedly, that models frequently output non-numerical responses in settings where there are no instructions to do so, or no in-context examples to follow.

On STS-B validation performance, models generally perform much better than on C-STS, with some models performing comparably to finetuned models. Since STS-B is included as a task in Natural Instructions v2 (Wang et al., 2022), it is likely to be recognizable to Flan-T5 models, which counts Natural Instructions v2 in its training data. Likewise, STS-B is comprised of long-existing and popular datasets, which plausibly exist in the the corpora used to train ChatGPT models.

**Processing Prompting Baseline Generations** For parsing prompting model generations, we allow for a maximum of 20 generation tokens. The output is stripped of non-numeric characters and errant punctuation before being cast to a float. For example, the response "The Answer is 2.0." is processed as 2.0 and counts as a valid prediction. If the cast fails, we mark the answer invalid and replace the predictions by a number $y \sim U[1, 5]$.

## B Prompt Examples

All prompts for the prompting baselines may consist of instructions, examples, and a query, though we include evaluations for no instructions and no examples in our results. Figure 5 shows an prompt example for the *short* instructions and $K = 2$ and

Figure 6 shows an example for *long* instructions and zero-shot setup.

**Instructions**

On a scale between 1 and 5, how similar are the following two sentences with respect to the condition provided? Respond only with a score between 1 and 5.

**Examples**

**Input: Sentence 1:** A bunch of blue buses parked in a parking lot in front of a housing community.
**Sentence 2:** Two buses, one blue and one red and white, are going to different destinations.
**Condition:** The type of transportation.
**Output:** 4.0
**Input: Sentence 1:** A bunch of blue buses parked in a parking lot in front of a housing community.
**Sentence 2:** Two buses, one blue and one red and white, are going to different destinations. Condition: The number of buses.
**Output:** 1.0

**Query**

**Input: Sentence 1:** The skater is descending the wooden wall beside the slope .
**Sentence 2:** A boy skateboards off a ramp covered in graffiti .
**Condition:** The location.
**Output:**

Figure 5: We show the full input for 2-shot setting with *short* instructions.

## C Crowdsourcing Guidelines

### C.1 Condition Annotation

We provide the complete condition annotation guidelines used for Mechanical Turk data collection in Figure 7.

### C.2 Condition Verification

We provide the complete verification guidelines used for Mechanical Turk data collection in Figure 8.

| Model | Modeling Type | Learning Rate | Weight Decay | Transform | Objective | Tri-Encoder Op. | Spearman | Pearson |
|---|---|---|---|---|---|---|---|---|
| RoBERTa$_{BASE}$ | Cross Encoder | 3.0e-05 | 0.10 | True | MSE | - | 41.02 | 40.95 |
| | Bi Encoder | 3.0e-05 | 0.10 | True | MSE | - | 37.93 | 37.17 |
| | Tri Encoder | 3.0e-05 | 0.00 | False | Quad + MSE | Concat | 28.70 | 27.50 |
| RoBERTa$_{LARGE}$ | Cross Encoder | 1.0e-05 | 0.10 | True | MSE | - | 40.21 | 40.49 |
| | Bi Encoder | 1.0e-05 | 0.10 | True | Quad + MSE | - | 35.81 | 33.25 |
| | Tri Encoder | 1.0e-05 | 0.00 | True | MSE | Hadamard | 21.82 | 21.46 |
| DiffCSE$_{BASE}$ | Cross Encoder | 3.0e-05 | 0.10 | False | MSE | - | 39.73 | 39.84 |
| | Bi Encoder | 3.0e-05 | 0.00 | False | MSE | - | 42.18 | 41.85 |
| | Tri Encoder | 3.0e-05 | 0.10 | False | Quad + MSE | Hadamard | 30.60 | 29.59 |
| SimCSE$_{BASE}$ | Cross Encoder | 3.0e-05 | 0.10 | True | MSE | - | 33.91 | 34.90 |
| | Bi Encoder | 3.0e-05 | 0.10 | False | MSE | - | 45.67 | 45.55 |
| | Tri Encoder | 3.0e-05 | 0.10 | False | Quad + MSE | Hadamard | 33.06 | 32.35 |
| SimCSE$_{LARGE}$ | Cross Encoder | 1.0e-05 | 0.10 | True | MSE | - | 44.31 | 44.42 |
| | Bi Encoder | 1.0e-05 | 0.10 | False | MSE | - | 47.70 | 47.41 |
| | Tri Encoder | 1.0e-05 | 0.00 | True | MSE | Hadamard | 34.46 | 34.95 |

Table 8: Fine-tuning models' results over *validation* split. We show the best performing configuration selected over the validation split which was the final configuration used to report each models' test performance.

| Model | Encoding | Learning Rate | Transform | Objective | Spearman | Pearson |
|---|---|---|---|---|---|---|
| RoBERTa$_{BASE}$ | Cross Encoder | 3.0e-05 | True | MSE | 90.54 | 90.55 |
| | Bi Encoder | 3.0e-05 | False | MSE | 87.23 | 86.73 |
| DiffCSE$_{BASE}$ | Cross Encoder | 3.0e-05 | False | MSE | 89.75 | 89.82 |
| | Bi Encoder | 3.0e-05 | False | MSE | 88.08 | 87.66 |
| RoBERTa$_{LARGE}$ | Cross Encoder | 3.0e-05 | True | MSE | 91.49 | 91.58 |
| | Bi Encoder | 3.0e-05 | False | MSE | 87.79 | 87.25 |
| SimCSE$_{BASE}$ | Cross Encoder | 3.0e-05 | True | MSE | 89.50 | 89.65 |
| | Bi Encoder | 3.0e-05 | False | MSE | 89.69 | 89.30 |
| SimCSE$_{LARGE}$ | Cross Encoder | 3.0e-05 | True | MSE | 91.73 | 91.78 |
| | Bi Encoder | 1.0e-05 | False | MSE | 90.70 | 90.56 |

Table 9: Validation performance of best sweep setting on STS-B.

| Data Ablation | Spear. | Pears. |
|---|---|---|
| Condition Only | 28.21 | 28.62 |
| Sentence Only | 9.98 | 9.51 |
| Baseline | 40.11 | 40.21 |

Table 10: When finetuned only with condition statement, RoBERTa$_{BASE}$ model can recover non-trivial performance, but falls well behind the baseline. Training on only the sentence pairs proves to be even less informative. We report the best validation performance over the same hyperparameter grid described in Section 4.1.

| Instruction | Model | 0-shot | | | 2-shot | | | 4-shot | | |
|---|---|---|---|---|---|---|---|---|---|---|
| | | Invalid | Pears. | Spear. | Invalid | Pears. | Spear. | Invalid | Pears. | Spear. |
| None | Flan-T5$_{\text{SMALL}}$ | 91.74 | 3.23 | 2.20 | 35.64 | 7.06 | 8.20 | 24.21 | 7.14 | 6.98 |
| | Flan-T5$_{\text{BASE}}$ | 97.18 | -4.25 | -3.65 | 6.42 | 5.51 | 9.86 | 2.40 | 11.39 | 12.11 |
| | Flan-T5$_{\text{LARGE}}$ | 98.69 | -2.86 | -1.47 | 13.37 | 13.27 | 13.26 | 2.68 | 13.98 | 12.74 |
| | Flan-T5$_{\text{XL}}$ | 86.27 | -0.81 | -0.69 | 8.29 | 11.21 | 12.81 | 0.53 | 18.15 | 17.05 |
| | Flan-T5$_{\text{XXL}}$ | 74.14 | 3.21 | 3.78 | 0.14 | 11.37 | 12.05 | 0.00 | 10.28 | 12.08 |
| | Flan-UL2 | 83.87 | 0.53 | 4.39 | 3.03 | 16.32 | 18.97 | 0.28 | 15.32 | 17.69 |
| | Tk-*Instruct*$_{3\text{B}}$ | 87.33 | -2.06 | -2.05 | 0.67 | 2.12 | 1.70 | 0.00 | 0.26 | 0.32 |
| | Tk-*Instruct*$_{11\text{B}}$ | 22.37 | 2.36 | 5.58 | 0.21 | 8.00 | 8.43 | 2.65 | 3.15 | 3.76 |
| | ChatGPT-3.5 | 65.24 | 5.80 | 11.21 | 17.57 | 3.96 | 3.91 | 2.43 | 6.49 | 6.31 |
| | GPT-4 | 59.17 | 9.01 | 16.69 | 4.98 | 16.10 | 15.56 | 0.60 | 26.74 | 26.59 |
| Short | Flan-T5$_{\text{SMALL}}$ | 0.00 | -5.02 | -7.43 | 0.00 | -4.99 | -5.81 | 0.00 | -4.24 | -4.29 |
| | Flan-T5$_{\text{BASE}}$ | 0.00 | 5.78 | 6.24 | 0.00 | 6.03 | 6.51 | 0.00 | 5.44 | 5.75 |
| | Flan-T5$_{\text{LARGE}}$ | 0.00 | 13.98 | 13.89 | 0.00 | 12.29 | 12.68 | 0.00 | 10.54 | 11.11 |
| | Flan-T5$_{\text{XL}}$ | 0.00 | 25.00 | 25.34 | 0.00 | 24.12 | 24.24 | 0.00 | 21.63 | 23.02 |
| | Flan-T5$_{\text{XXL}}$ | 0.00 | 29.95 | 29.50 | 0.00 | 29.69 | 30.39 | 0.00 | 28.95 | 29.49 |
| | Flan-UL2 | 0.00 | 20.83 | 20.24 | 0.00 | 21.98 | 21.70 | 0.00 | 22.52 | 22.62 |
| | Tk-*Instruct*$_{3\text{B}}$ | 76.36 | 0.34 | 1.62 | 0.04 | 5.23 | 5.19 | 0.00 | 2.80 | 2.68 |
| | Tk-*Instruct*$_{11\text{B}}$ | 0.04 | 9.50 | 11.78 | 0.04 | 22.10 | 23.84 | 0.00 | 15.65 | 17.56 |
| | ChatGPT-3.5 | 0.00 | 12.91 | 11.13 | 0.04 | 16.63 | 17.62 | 0.07 | 12.60 | 13.76 |
| | GPT-4 | 0.00 | 38.77 | 39.47 | 0.00 | 39.76 | 41.25 | 0.00 | 41.52 | 42.05 |
| Long | Flan-T5$_{\text{SMALL}}$ | 0.00 | -1.48 | -1.58 | 0.00 | -2.71 | -3.14 | 0.00 | -4.80 | -4.67 |
| | Flan-T5$_{\text{BASE}}$ | 0.00 | 6.53 | 6.08 | 0.00 | 10.87 | 10.44 | 0.00 | 8.47 | 7.59 |
| | Flan-T5$_{\text{LARGE}}$ | 0.00 | 11.21 | 10.64 | 0.00 | 11.37 | 11.08 | 0.00 | 10.68 | 10.25 |
| | Flan-T5$_{\text{XL}}$ | 0.00 | 24.97 | 25.01 | 0.00 | 23.76 | 23.86 | 0.00 | 23.59 | 23.72 |
| | Flan-T5$_{\text{XXL}}$ | 0.00 | 29.71 | 29.79 | 0.00 | 30.68 | 30.69 | 0.00 | 30.01 | 29.99 |
| | Flan-UL2 | 0.00 | 21.27 | 21.07 | 0.00 | 22.08 | 21.64 | 0.00 | 21.91 | 21.56 |
| | Tk-*Instruct*$_{3\text{B}}$ | 0.14 | 2.10 | 1.88 | 0.00 | 4.29 | 3.84 | 0.00 | 0.95 | 0.93 |
| | Tk-*Instruct*$_{11\text{B}}$ | 0.00 | 9.24 | 11.24 | 0.00 | 19.82 | 21.23 | 0.00 | 16.06 | 17.38 |
| | ChatGPT-3.5 | 0.00 | 10.24 | 8.42 | 0.00 | 16.82 | 15.46 | 0.00 | 16.60 | 15.70 |
| | GPT-4 | 0.00 | 33.48 | 33.04 | 0.00 | 39.08 | 39.53 | 0.00 | 42.26 | 42.38 |

Table 11: Validation performance for prompting baselines on C-STS.

| Instruction | Model | 0-shot | | | 2-shot | | | 4-shot | | |
|---|---|---|---|---|---|---|---|---|---|---|
| | | Invalid | Pears. | Spear. | Invalid | Pears. | Spear. | Invalid | Pears. | Spear. |
| None | Flan-T5$_{\text{SMALL}}$ | 89.20 | 0.53 | 0.80 | 48.00 | -0.90 | -3.26 | 46.87 | -2.00 | 4.21 |
| | Flan-T5$_{\text{BASE}}$ | 92.87 | -1.38 | 4.03 | 3.67 | 0.25 | 41.81 | 3.67 | 40.71 | 39.76 |
| | Flan-T5$_{\text{LARGE}}$ | 90.07 | -1.06 | 5.64 | 9.67 | 5.29 | 65.38 | 2.67 | 67.44 | 68.90 |
| | Flan-T5$_{\text{XL}}$ | 87.53 | -1.71 | -0.96 | 3.80 | 69.87 | 73.70 | 0.73 | 72.77 | 76.41 |
| | Flan-T5$_{\text{XXL}}$ | 63.80 | -4.34 | 13.41 | 0.00 | 65.44 | 67.50 | 0.00 | 70.87 | 71.60 |
| | Flan-UL2 | 97.00 | -0.41 | 2.74 | 6.60 | 73.17 | 75.51 | 1.53 | 80.01 | 81.66 |
| | Tk-*Instruct*$_{\text{3B}}$ | 69.20 | 3.90 | 5.22 | 0.07 | 8.04 | 7.97 | 0.27 | 6.65 | 8.34 |
| | Tk-*Instruct*$_{\text{11B}}$ | 2.87 | 3.39 | 8.71 | 0.13 | 5.43 | 9.88 | 0.13 | 11.65 | 15.47 |
| | ChatGPT-3.5 | 96.93 | -4.17 | 1.61 | 0.07 | 63.86 | 64.83 | 0.00 | 74.96 | 76.15 |
| | GPT-4 | 63.20 | -2.40 | 20.10 | 0.00 | 76.70 | 75.92 | 0.00 | 86.16 | 86.25 |
| Short | Flan-T5$_{\text{SMALL}}$ | 0.07 | 18.44 | 18.43 | 0.07 | 19.09 | 19.21 | 0.00 | 19.97 | 20.41 |
| | Flan-T5$_{\text{BASE}}$ | 0.00 | 80.98 | 80.94 | 0.00 | 80.91 | 80.97 | 0.00 | 81.27 | 81.31 |
| | Flan-T5$_{\text{LARGE}}$ | 0.00 | 87.85 | 87.89 | 0.00 | 86.90 | 87.34 | 0.00 | 86.44 | 86.88 |
| | Flan-T5$_{\text{XL}}$ | 0.00 | 89.69 | 89.76 | 0.00 | 89.57 | 89.48 | 0.00 | 89.53 | 89.36 |
| | Flan-T5$_{\text{XXL}}$ | 0.00 | 89.80 | 89.79 | 0.00 | 88.33 | 88.62 | 0.00 | 86.54 | 87.38 |
| | Flan-UL2 | 0.00 | 91.57 | 91.62 | 0.00 | 91.72 | 91.62 | 0.00 | 91.60 | 91.48 |
| | Tk-*Instruct*$_{\text{3B}}$ | 63.93 | 5.00 | 20.17 | 0.00 | 49.86 | 51.24 | 0.00 | 48.08 | 48.58 |
| | Tk-*Instruct*$_{\text{11B}}$ | 0.07 | 35.04 | 34.79 | 0.00 | 50.65 | 54.01 | 0.00 | 48.48 | 51.99 |
| | ChatGPT-3.5 | 0.00 | 86.58 | 86.78 | 0.00 | 83.69 | 83.13 | 0.00 | 85.12 | 84.91 |
| | GPT-4 | 0.00 | 88.20 | 88.95 | 0.00 | 88.38 | 88.44 | 0.00 | 89.02 | 88.96 |
| Long | Flan-T5$_{\text{SMALL}}$ | 0.00 | 6.33 | 5.98 | 0.00 | 14.92 | 14.75 | 0.00 | 16.81 | 16.06 |
| | Flan-T5$_{\text{BASE}}$ | 0.00 | 82.32 | 82.22 | 0.00 | 82.42 | 82.49 | 0.00 | 82.14 | 82.16 |
| | Flan-T5$_{\text{LARGE}}$ | 0.00 | 89.81 | 89.86 | 0.00 | 89.85 | 89.85 | 0.00 | 89.55 | 89.56 |
| | Flan-T5$_{\text{XL}}$ | 0.00 | 90.33 | 90.62 | 0.00 | 89.86 | 90.12 | 0.00 | 90.43 | 90.66 |
| | Flan-T5$_{\text{XXL}}$ | 0.00 | 90.75 | 90.97 | 0.00 | 91.58 | 91.51 | 0.00 | 91.50 | 91.35 |
| | Flan-UL2 | 0.00 | 91.02 | 91.68 | 0.00 | 91.45 | 91.90 | 0.00 | 91.67 | 92.05 |
| | Tk-*Instruct*$_{\text{3B}}$ | 0.00 | 23.90 | 26.76 | 0.00 | 66.89 | 67.63 | 0.00 | 65.19 | 66.04 |
| | Tk-*Instruct*$_{\text{11B}}$ | 0.00 | 64.09 | 65.20 | 0.00 | 67.93 | 69.06 | 0.00 | 61.38 | 63.65 |
| | ChatGPT-3.5 | 0.00 | 86.28 | 86.59 | 0.00 | 86.16 | 85.81 | 0.00 | 87.08 | 86.90 |
| | GPT-4 | 0.00 | 89.57 | 89.76 | 0.00 | 90.01 | 89.95 | 0.00 | 90.73 | 90.65 |

Table 12: Validation performance for prompting baselines on STS-B.

**Definition:** Evaluate the similarity between the two sentences, with respect to the condition. Assign the pair a score between 1 and 5 as follows:

1 : The two sentences are completely dissimilar with respect to the condition.

2 : The two sentences are dissimilar, but are on a similar topic with respect to the condition.

3 : The two sentences are roughly equivalent, but some important information differs or is missing with respect to the condition.

4 : The two sentences are mostly equivalent, but some unimportant details differ with respect to the condition.

5 : The two sentences are completely equivalent with respect to the condition.

**Input: Sentence 1**: Elderly man sitting on a blue couch reading a paper.
**Sentence 2:** Older man riding public transportation while reading a newspaper.
**Condition:** The location of the man.
**Output:**

Figure 6: The full text input for the zero-shot evaluation with large language models, using 'long' instructions. Emphasis and section titles added for clarity.

## Task summary

Our goal is to understand the similarity of a sentence pair based on a condition.
Concretely, for a sentence pair (S1 and S2), provide one condition (C-High) such that S1 and S2 have high similarity, and one condition (C-Low) such that they have low similarity.

As an example:
> **S1**: A large green ball was bouncing on the street.
> **S2**: I bought a small green avocado.

> **C-High**: The color (High Similarity because it is green in both sentences)
> **C-Low**: The size (Low Similarity because it is large in the first and small in the second sentence)

Conditions are English phrases which are used to choose an aspect of the sentence.

## Guidelines for conditions

You are allowed to use the internet to understand the sentences, but the conditions need to be written by you. The following guidelines need to be followed.

1. **Conditions should be grammatically correct** English phrases or sentences.
2. **Avoid conditions which reference missing information** that cannot be inferred from sentences. For example, avoid the following condition, because the color of the animal in S2 is unclear.
   a. S1: Brown bears attacked people in the night.
   b. S2: Some dogs were barking on the road.
   c. C-High: The color of the animal.
   But the following is a good condition because it can be inferred that the game is chess:
   d. S1: Black ultimately reached an endgame two pawns up.
   e. S2: Now the white king comes just in time to support his pawn.
   f. C-High: The game being played.
3. **Conditions should reference aspects or attributes of sentences and not the values.** For example, the following ("The color is green") is an incorrect condition because it directly mentions "green", which is the value of the attribute "color":
   a. S1: A large green watermelon.

Figure 7: Annotation Guidelines

b.  S2: A green avocado in the basket.
c.  C-High: The color is green.
Instead, the same condition can correctly be written as: "The color of the fruit".
4.  **Avoid** **conditions which explicitly use words like "sentences"**. For example, instead of saying "the color in the sentence", just say "The color".
5.  **Avoid** **vague conditions which do not help narrow down a specific aspect of the sentence**. For example, avoid conditions which simply say "The activity", which does not help narrow down the aspect. Instead use more informative words like "the sport" or "the hobby" as much as possible.
6.  **Whenever possible, try to write conditions which refer to abstract similarity**.
Consider the following sentences:
a.  Two women are celebrating a goal.
b.  A couple is eating a tasty meal.
A condition which is more abstract is preferred:
c.  *Abstract condition*: The sentiment of the people.
Although a more literal condition is valid, it is less preferred:
d.  *Literal condition*: The number of people.

## Examples

We provide good and bad examples of conditions for sentence pairs, along with the reasoning.

### Good examples

All the following conditions are valid because they follow our guidelines.

| Sentence 1 | Sentence 2 | Condition | Similarity | Explanation |
|---|---|---|---|---|
| The moon looked incredible! | The car was completely covered in snow. | The color of the object. | High | The color is white in both cases. This is a good condition because it references the color of the object without explicitly mentioning it. |
| A group of people wearing helmets and riding on bikes. | A group of bikers are gathered together and taking pictures. | The speed of the cyclists. | Low | The group of cyclists is moving in the first sentence whereas they are not in the second. Hence their speeds are dissimilar. |
| Three people are holding a ladder while another climbs it. | Three people are listening to music in a car. | The number of people. | Low | There are four people in the first sentence but only three in the second sentence. |

**Bad examples**

All the following conditions are invalid because they ignore one or more of our guidelines.

| Sentence 1 | Sentence 2 | Condition | Reason for invalidity of condition |
|---|---|---|---|
| Egyptians appeased gods with offerings and prayers. | People in this era put faith in specific gods to protect their lives. | The culture involved. | It violates guideline 2.
The culture in the second sentence cannot be inferred and is missing information. |
| An adult elephant is playing in the river. | A boulder is rolling down the hill. | The size of the object is large. | It violates guideline 3.
The condition should have been "The size of the object", without explicitly referring to it being "large". |
| A guitarist is playing on a bench. | A man in a green hat is playing the guitar on the road. | The instrument in the sentence. | It violates guideline 4.
The condition would be good if "in the sentence" was removed so that it is just "The instrument". |
| A middle-aged man is helping construct a grass hut. | Three men work on a roof. | The activity. | This condition is too vague and does not reference a specific aspect. A better condition would be: "The type of construction". |
| A man on top of a partially completed roof laying down more shingles. | A man in a hard hat and safety gear stands in a construction site. | The number of people. | While this condition is valid, it violates guideline 6, which says that an abstract condition should be considered wherever possible.
A better condition would have been, "*The occupation of the man*", which is "construction worker" in both cases. |

## Task summary

Our goal is to understand the similarity of a sentence pair based on a condition.
Concretely, for a sentence pair (**S1** and **S2**), and a condition '**C**', provide a score which indicates the similarity of **S1** and **S2** with respect to **C**.

As an example:
> **S1**: A large green ball was bouncing on the street.
> **S2**: I bought a small green avocado.
> **C**: The size of the object

> **Similarity**: 1 (Low Similarity because it is large in the first and small in the second sentence)

## Guidelines for annotating similarity

### Part 1

Given two sentences and a condition, **first check if the condition applies to both the sentences**. If the condition does not apply even to one of the sentences, please check the box provided to indicate the same. For example:
> **S1**: A small dog happily runs across the street.
> **S2**: I bought a small green avocado.
> **C**: The sentiment

In the above example, the condition does not make sense for S2 because there is no sentiment that can be inferred from it.

### Part 2

If the condition makes sense, given two sentences and a condition, please ascribe a similarity score for the sentences when interpreted with respect to the condition.
**The score has to be one of the following numbers: {1, 2, 3, 4, 5}.**
Tips:
- **Please avoid overusing the middle range score (3) as much as possible.**
- **Feel free to use the extreme scores (1 and 5) if they make sense to you.**

The following is the meaning of the different scores:
1. **Score = 1**: *The two sentences are completely dissimilar with respect to the condition.*
   For example:

Figure 8: Verification Guidelines

   **S1**: A man cooks in the kitchen.

   **S2**: A woman is riding a bike on the road.

   **C**: The gender (Man and woman are dissimilar with respect to gender)

2. **Score = 2**: *The two sentences are dissimilar, but are on a similar topic with respect to the condition.*

 For example:

   **S1**: A man plays the guitar.

   **S2**: A little girl listens to the violin.

   **C**: The instrument (Both are string instruments, similar but different instruments)

3. **Score = 3**: *The two sentences are roughly equivalent, but some important information differs or is missing with respect to the condition.*

 For example:

   **S1**: A small crowd gathered around the injured person.

   **S2**: A crowd jumps up and down to the tunes played by an artist.

   **C**: Number of people

   (While both are crowds, it is important and unclear how many people there are.)

4. **Score = 4**: *The two sentences are mostly equivalent, but some unimportant details differ with respect to the condition.*

 For example:

   **S1**: The little girl plays the jazz guitar.

   **S2**: The guitar looked nice and shiny.

   **C**: The instrument (Guitar in both cases, but the exact type is different and unimportant)

5. **Score = 5**: *The two sentences are completely equivalent as they mean the same thing with respect to the condition.*

 For example:

   **S1**: Three boys play on the playground.

   **S2**: There are 3 girls near the fountain.

   **C**: The number of people (3 and three are strictly equivalent)