# OpenReview forum: "C-STS: Conditional Semantic Textual Similarity"
_EMNLP/2023/Conference — EMNLP 2023 Main_

### Official Review · Reviewer_L1HG · 2023-07-26

**Soundness:** 4

**Excitement:**

4: Strong: This paper deepens the understanding of some phenomenon or lowers the barriers to an existing research direction.

**Missing References:**

None that I am aware of, although I am not very familiar with the STS literature.

**Paper Topic And Main Contributions:**

This paper introduces a novel task, *conditional* semantic textual similarity (C-STS). The well-established semantic textual similarity (STS) task requires systems to assign a scalar value $y$ to a pair of sentences $(s_1,s_2)$ indicating the degree of semantic similarity between $s_1$ and $s_2$. C-STS modifies this task definition by soliciting semantic similarity judgments relative to a particular *condition*, $c$, which expresses some feature or aspect with respect to which the semantic similarity of $s_1$ and $s_2$ should be evaluated. The paper presents a 19K-example dataset of $(s_1,s_2,c,y)$ tuples, where $s_1$ and $s_2$ are image captions drawn from existing image captioning datasets, $c$ is a human-written condition, and $y$ is a human-annotated similarity judgment on $s_1$ and $s_2$ relative to $c$. Evaluations across a variety of different models and settings (zero-shot, few-shot, and fine-tuned) demonstrate that C-STS is challenging even for very recent, top-performing models.

**Questions For The Authors:**

A. Given the step in the data collection procedure in which annotators are explicitly asked to generate both a high similarity and a low similarity condition for each sentence pair, I'm somewhat puzzled as to why the distribution in Figure 4 isn't more balanced between low and high similarity judgments (e.g. it seems like there are almost twice as many 4 and 5 responses (each) as there are 1 responses).


B. There is a fair amount of past work at this point, primarily in the NLI literature (e.g. [here](https://aclanthology.org/Q19-1043/) and [here](https://aclanthology.org/2020.starsem-1.9.pdf)) that shows that individual annotators' response distributions (e.g. on a Likert scale) can vary considerably (e.g. some people will tend to cluster their responses in the 2-4 range, while others might tend to use more 1s and 5s), likely even when given criteria for each rating or admonitions to use the full scale like the ones provided in the appendix of this paper. This seems like a potential concern here, given that there is no redundancy in the annotation (except on the 100-example subset used for IAA evaluation, unless I'm mistaken). I would be curious to know if the authors plan to release (anonymized) annotator information along with each example for researchers interested in modeling this variability.

C. A relatively minor point: the train/validation/test split ratio allocates fewer examples to train than I'm accustomed to seeing (I much more commonly see 80/10/10 or at least 70/15/15; perhaps this is an idiosyncratic feature of my research background). Yet, L252-3 says this ratio is "standard practice." If this is so, it would be good to provide references; if not, it would be good to know how this ratio was decided.

**Reasons To Accept:**

- The C-STS task seems well-motivated and potentially of interest to a broad subset of the NLP community, including the IR, NLI, and RTE communities, as well as folks working at the intersection of vision and language.
- The experiments are quite thorough while leaving ample room for future work, and substantial details about both the data collection procedure and experimental setup are provided in the appendices.


**Reasons To Reject:**

I do not have strong reasons to reject, but a number of more modest concerns, which I raise in the section below.

**Reproducibility:**

4: Could mostly reproduce the results, but there may be some variation because of sample variance or minor variations in their interpretation of the protocol or method.

**Reviewer Confidence:**

3: Pretty sure, but there's a chance I missed something. Although I have a good feel for this area in general, I did not carefully check the paper's details, e.g., the math, experimental design, or novelty.

**Typos Grammar Style And Presentation Improvements:**

Typos
- L232: missing space between "pair" and "(Appendix C.1)"
- L317: "for" -> "with"
- L544: "corroborates with" -> "corroborates"

Presentation
- Given that the judgments are ordinal, Kendall's $\tau$ or Krippendorff's $\alpha$ seem like more appropriate measures of IAA than Fleiss's $\kappa$.
- Section 7 is more of a "conclusion" section than a "discussion" section; it might be advisable to rename it to the former.
- The ethics statement appears to be missing.
- The principle item in the limitations section is that the dataset could be bigger. This is a fair point, but could surely be said of most datasets. It seems like there are potentially additional, more important limitations that others should be aware of (e.g. the apparent skew in the response distribution I raise above).

---

> ### Author Rebuttal · Authors · 2023-08-28
>
> Thank you for your thorough review, we greatly appreciate your feedback and comments.
>
> *A. Given the step in the data collection procedure in which annotators are explicitly asked to generate both a high similarity and a low similarity condition for each sentence pair, I'm somewhat puzzled as to why the distribution in Figure 4 isn't more balanced between low and high similarity judgments (e.g. it seems like there are almost twice as many 4 and 5 responses (each) as there are 1 responses).*
>
> We perform our annotation process in two steps (1) acquiring the C-High and C-Low conditions and (2) acquiring similarity judgements. Step 1 is done without explicit reference to a Likert scale or ordinal rating system, we simply ask for a high and low condition provided some examples. During step 2, similarity judgements are not primed with whether the condition was acquired as high or low and human annotators’ Likert rating may be slightly biased towards higher similarity judgements. Additionally, during step 2, we always have annotators rate whether the condition is high-quality and applicable to both sentences. We find that annotators are more likely to eliminate low-similarity conditions in this phase, which also results in instance pairs with slightly higher-skewed similarity scores. Overall, this results seems to be the result of our straightforward annotation process and natural annotator perceptions. You can see further details on the annotator guidelines in Appendix C.
>
> However, our quality assessment and inter annotator agreement scores can confirm that the quality of the dataset is high.
>
>
> *B. There is a fair amount of past work at this point, primarily in the NLI literature (e.g. here and here) that shows that individual annotators' response distributions (e.g. on a Likert scale) can vary considerably (e.g. some people will tend to cluster their responses in the 2-4 range, while others might tend to use more 1s and 5s), likely even when given criteria for each rating or admonitions to use the full scale like the ones provided in the appendix of this paper. This seems like a potential concern here, given that there is no redundancy in the annotation (except on the 100-example subset used for IAA evaluation, unless I'm mistaken). I would be curious to know if the authors plan to release (anonymized) annotator information along with each example for researchers interested in modeling this variability.*
>
> Like you pointed out, there is in fact redundancy when measuring the IAA, but the redundancy does not naturally exists for the rest of the data. But the additional motivation of our work was indeed to avoid the 2-4 gravitation that annotators naturally get into.
> But just to make sure we understand correctly, could you elaborate on what kind of anonymized annotator information you are looking for? Are they demographics related?
>
>
> *C. A relatively minor point: the train/validation/test split ratio allocates fewer examples to train than I'm accustomed to seeing (I much more commonly see 80/10/10 or at least 70/15/15; perhaps this is an idiosyncratic feature of my research background). Yet, L252-3 says this ratio is "standard practice." If this is so, it would be good to provide references; if not, it would be good to know how this ratio was decided.*
>
> Regarding C) our standard practice referred to having a significantly larger training set that the eval and test sets. The reason we went for a 60:15:25 split is that we wanted to thoroughly test the model with a large amount of data rather than provide a large training set and test on a small test set. This is because C-STS is a novel task and we believe that the community wil benefit from a thorough evaluation. We will change the writing to reflect our motivation. Thank you for pointing this out.
>
> *Given that the judgments are ordinal, Kendall's tau or Krippendorff's alpha seem like more appropriate measures of IAA than Fleiss's k.*
>
> Thanks for this suggestion, we will incorporate this in the camera-ready version by providing both the judgements.
>
>
> *Section 7 is more of a "conclusion" section than a "discussion" section; it might be advisable to rename it to the former.*
>
> We will rename this accordingly.
>
> We’d also like to acknowledge your other comments/corrections and we shall be updating the final paper to address each of your suggestions. Thank you again for your time and feedback.

---

### Official Review · Reviewer_4jPR · 2023-07-30

**Soundness:** 4

**Excitement:**

4: Strong: This paper deepens the understanding of some phenomenon or lowers the barriers to an existing research direction.

**Paper Topic And Main Contributions:**

This paper proposes a new task called conditional sentence similarity assigning similarity scores to a pair of sentences and a condition sentence. The condition sentence makes the task less ambiguous and requires the model to capture more fine-grained semantic signals from the given two sentences. The experiments are conducted with a nice set of baselines: conventional fine-tuning-based methods with encoder-based models and prompting-based methods with fine-tuned and proprietary LLMs.
The paper is well-written and easy to follow.

**Reasons To Accept:**

* The paper proposes a new benchmark useful for evaluating the language models
* The paper contains a nice set of baselines

**Reasons To Reject:**

N/A

**Reproducibility:**

4: Could mostly reproduce the results, but there may be some variation because of sample variance or minor variations in their interpretation of the protocol or method.

**Reviewer Confidence:**

3: Pretty sure, but there's a chance I missed something. Although I have a good feel for this area in general, I did not carefully check the paper's details, e.g., the math, experimental design, or novelty.

---

> ### Author Rebuttal · Authors · 2023-08-28
>
> We thank you for taking the time to read our paper.

---

### Official Review · Reviewer_cVEM · 2023-08-04

**Soundness:** 4

**Excitement:**

4: Strong: This paper deepens the understanding of some phenomenon or lowers the barriers to an existing research direction.

**Paper Topic And Main Contributions:**

The authors propose a novel sentence similarity task (called conditional semantic textual similarity  C-STS) that resolves the problem of the inherent ambiguity in semantic textual similarity (STS). The authors create the C-STS-2023 dataset to test models on C-STS. C-STS-2023 dataset includes instances containing sentence pairs, a condition, and a scalar similarity judgement.

**Reasons To Accept:**

The paper can be accepted for the following reasons: the authors offer a new approach to measure a sentence similarity. The title and abstract are correctly selected and written. The paper is correctly structured, easy to follow. Pictures and tables are relevant and clearly explained.

**Reasons To Reject:**

As an option the size of the dataset can be increase to ensure sentence embedding style models have additional data for fine-tuning.

**Reproducibility:**

5: Could easily reproduce the results.

**Reviewer Confidence:**

3: Pretty sure, but there's a chance I missed something. Although I have a good feel for this area in general, I did not carefully check the paper's details, e.g., the math, experimental design, or novelty.

---

> ### Author Rebuttal · Authors · 2023-08-28
>
> We thank you for taking the time to read our paper and give useful comments.
> We believe it is a good suggestion to increase the size of the dataset and are working on semi-automated ways to collect a large dataset without significantly increased human effort.

---

### Meta-Review · Area_Chair_GMVq · 2023-09-20

**Recommendation:** 5

**Metareview:**

This paper presents a novel task and dataset, enhancing the semantic textual similarity (STS) by introducing conditional elements. Even in the era of large-scale language models, predicting textual similarity remain vital, with applications like retrieval-augmented generation. We look forward a refined camera-ready version and updated dataset that takes into account feedback from reviews and discussions with reviewer L1HG.

---

### Decision · Program_Chairs · 2023-10-07

**Decision:**

Accept-Main

**Comment:**

This paper presents a novel task and dataset, enhancing the semantic textual similarity (STS) by introducing conditional elements. Even in the era of large-scale language models, predicting textual similarity remain vital, with applications like retrieval-augmented generation. We look forward a refined camera-ready version and updated dataset that takes into account feedback from reviews and discussions with reviewer L1HG.